# Natural Occurring Terpene Cyclic Anhydrides: Biosynthetic Origin and Biological Activities

**DOI:** 10.3390/biom14080955

**Published:** 2024-08-06

**Authors:** Diego O. Molina Inzunza, Juan E. Martín González, María José Segura Navarro, Alejandro F. Barrero, José F. Quílez del Moral

**Affiliations:** Department of Organic Chemistry, Institute of Biotechnology, University of Granada, 18071 Granada, Spain; diegomolina96@gmail.com (D.O.M.I.); juanmartingonzalez.jmn@gmail.com (J.E.M.G.); mariajseguranavarro@gmail.com (M.J.S.N.)

**Keywords:** natural product, anhydrides, biosynthesis, biological activity

## Abstract

Cyclic acid anhydride is a not very widespread structure in nature, but with a determining role in natural products possessing this functionality in their skeleton. To the best of our knowledge, no revision of terpenes containing cyclic anhydrides has been previously reported. The result was that more than 100 terpenic cyclic anhydrides and related compounds were found to be in need of being reported. This review has been systematically organized by terpene skeletons, from the smallest to largest, describing their sources and bioactivities. In addition, different biosynthetic pathways for their final oxidations, namely, routes A, B and C, leading to the formation of these heterocyclic natural products, have been proposed. We have also included the most plausible precursors of these natural products, which mostly happened to be present in the same natural source. Some molecules derived from terpene cyclic anhydrides, such as their natural imide derivatives, have also been described due to their significant biological activity. In this sense, special attention has been paid to cantharidin because of its historical relevance and its broad bioactivity. A plausible biosynthesis of cantharidin has been proposed for the first time. Finally, cyclic anhydride structures that were firstly assigned as anhydrides and later corrected have been also described.

## 1. Introduction

Molecules of natural origin containing in their structure the functional grouping cyclic acid anhydride, also called cyclic anhydride, have aroused interest for their structural uniqueness and, in many cases, for their properties. This function is present in several biosynthetic families of natural products, especially terpenes. The importance of these molecules lies in the fact that the anhydride function is recognized as a strong electrophile, having a high acylating capacity [1,2], and on the other hand it many possess interesting biological activities.

Natural products possessing cyclic anhydrides are present in all the kingdoms of living beings, which generally use them as a defense or attack against predators and biological competitors. Thus, their bioactivity covers a wide spectrum, highlighting, among others, cytotoxic, biocide, purgative or sexual enhancer roles, etc. Because of this, they have gained interest at the industrial and pharmacological level, as a natural alternative to synthetic drugs and phytosanitary products.

The present work has the objective of compiling, in a coherent and ordered way, those terpenes of natural origin that contain the cyclic anhydride grouping in their structure. For this purpose, their presence in different organisms, biosynthetic origin, biological activities and applications will be considered. In addition, terpenes possessing functions that are considered immediate precursors of anhydrides, and other direct derivatives such as imides, will be collected.

To approach this review, we started with CAS Scifinder, where we performed a keyword search using “cyclic anhydride” and “cyclic acid anhydride” to obtain more than 10,000 results. Filtering searches that included the terms “natural product”, “terpene”, “terpenic”, “terpenoids”, “monoterpene/monoterpenoids”, “sesquiterpene/sesquiterpenoids” etc., did not prove to be useful, and few publications meeting the objective of this review were found. We then focused our search on chemical structure, exploring different sizes of cyclic anhydrides, that is, 5-, 6-, and 7-membered rings, which provided more than 40,000 results among them. To discard molecules of non-natural origin, we filtered the search by excluding structures containing metals, more than one compound, etc. In addition, to facilitate the search, we sorted the results by molecular size to finally find the herein collected structures.

Review of other databases, such The Natural Products Atlas or COCONUT Database, did not lead to the identification of new molecules [3,4].

From an initial number of results in the tens of thousands, more than eighty cyclic terpenic anhydrides and more than forty related imides have been found.

Cyclic anhydride terpenes present a great variability in their skeletons, since terpenoids constitute the most numerous and structurally diverse families of natural products. This is a consequence of the biosynthetic processes of formation that take place in several successive stages, their ubiquity in nature and biological specificity [5] The biosynthesis of these compounds has been represented here in a summarized form in four phases, I–IV (Figure 1), considering the structural changes that occur and the enzymes responsible for them [6,7].

Phase I begins with the synthesis of the universal C5 terpene precursors isopentenyl diphosphate (IPP) and its isomer dimethylallyl diphosphate (DMAPP) (Figure 1). Pathways known as mevalonate (MVA) and methyl-erythritol 4-phosphate (MEP) are responsible for the synthesis of these building blocks [8,9,10,11]. In phase II, the formation of each of the acyclic prenyldiphosphates (PDPs) with different molecular size multiples of C5, takes place by condensation of one or several IPPs and DMAPPs and is catalyzed by prenyltransferases. This sequential elongation mostly complies with the so-called isoprene rule [12]. In phase III, the PDPs are transformed into a backbone of large diversity and complexity by the catalysis of terpene synthases (TPSs) and cyclases (TCSs) [13]. Phase IV is characterized by multiple changes in the organic functions of each terpene, mainly oxidation–reduction, and sometimes in the main backbone (cleavages, rearrangements, C-C bond formation). These processes are catalyzed in most cases by cytochrome P450-dependent oxygenases, monooxygenases and dioxygenases. This is the phase in which terpenoids possessing cyclic anhydrides originate, as well as highly functionalized terpenoids [14,15,16]. In this review, within phase IV, three different biosynthetic routes (routes A–C, Figure 1) leading to the formation of the cyclic anhydride cluster in terpenes have been considered. Each route is characterized by having a common precursor (structure or functions leading to the cyclic anhydride), and consequently, for each route the same transformations catalyzed by enzymes of the same nature will be repeated.

Oxidation is the chemical reaction that nature uses to generate a wide range of compounds from a single molecular backbone. Monooxygenases and dioxygenases are the enzymes involved in these chemical changes, and they are very versatile, being able to act once or several times on the same and different positions of the structure. A very high number of reactions in the biogenetic oxidation of terpenes are monooxygenation catalyzed by cytochromes P450 (P450s), a large family of “iron and heme-dependent monooxygenases”, and far from being the only ones, we also find oxidations mediated by nonheme iron-dependent oxygenases, the new radical superfamily *S*-adenosylmethionine (SAM), copper-dependent oxygenases (cytochrome c oxidase, laccase and tyrosinase), flavin-dependent monooxygenase (FMOs) or NAD(P)H-dependent reductases [17]. On the other hand, dioxygenases are constituted by several families of enzymes, the most relevant being lipoxygenases (LOXs), 2-oxoacid-dependent dioxygenases (2-ODDs) and diol-cleaving catechol dioxygenases. The last two are the most important in the biosynthesis of terpenic anhydrides [18,19]. This process is not exclusive to these enzymes and can also be catalyzed by biomimetic or bioinspired chemical reactions, which mimic, partially or totally, the action of the oxidase [20,21,22,23,24].

## 2. Final Oxidation Pathways and Formation of Terpenic Cyclic Anhydrides

The three main biosynthetic pathways, A–C, of phase IV towards the biosynthetic formation of cyclic terpenic anhydrides are summarized below, commenting on the key steps of each mechanism involved. To illustrate each of them, an example of the biosynthesis of a specific terpene is provided.

### 2.1. Route Type A

Pathway A, the most common and the one that generates most of these terpenic anhydrides, is the consequence of the successive and, if necessary, multiple oxidations of one or two methyls that are spatially close to each other in the carbon skeleton. The progressive oxidation of each one passes through alcohol and aldehyde to carboxylic acids (Figure 2). These reactions are carried out by the repeated action of monooxygenases on the same methyl. It is not necessary that the oxidation processes are simultaneous for both methyls, or that they act on both. They usually are not, so we define the pathway as the oxidation of methyls, hydroxymethyls or aldehydes to carboxylic acids, depending on which substrate we take as a starting point. The cyclic terpenic anhydride can be found in equilibrium with the carboxylic acid, which will be displaced to a greater or lesser extent towards one or the other function depending on the conditions (the pH of the medium, temperature, etc.). Spatial arrangement and angular tension permitting, the cyclic acid anhydride is usually favored.

*Seco*iridoid anhydride **I**, isolated from the plant *Stachys viticina*, represents a clear example of a cyclic anhydride terpene that is formed following the type A pathway [25]. In this regard, chrysomelidial (**II**), an iridoid dialdehyde acting as a defensive toxin, has been isolated in the larvae of the beetle *Phaedon cochleariae* [26]. It was proposed that chrysomelidial (**II**) is a plausible intermediate in the formation of anhydride **I** and therefore, the biosynthetic pathway leading to both molecules should be analogous. Once generated, the chrysomelidial (**II**) compound undergoes two oxidations, one at each aldehyde being catalyzed by monooxygenases. The resulting carboxylic acid evolves by dehydration into anhydride **I** (Figure 1).

### 2.2. Route Type B

Route B is the pathway followed for the biosynthesis of a significant number of natural terpene anhydrides. This route is characterized by the oxidative breakage of carbon–carbon bonds, catalyzed by monooxygenases or dioxygenases. Thus, terpene anhydrides originate by degradation from various functional groups such as olefins, enols, 1,2-diols, catechols, alpha-hydroxycarbonyls and 1,2- and 1,3-dicarbonyls. Fragmentation is marked by the specific functional group and the enzyme catalyzing the oxidation. The most relevant mechanisms have been classified into two classes.

#### 2.2.1. Route B1

B1 occurs with the intermediate formation of dioxetanes, is used for the C-C double bond of olefins, enols, catechols, etc., and is catalyzed by dioxygenases. Intermediate dioxetane evolves rapidly by cycloreversion to dialdehyde, aldehyde + acid or diacid; in the first two cases, the aldehydes are further oxidized by monooxygenases to di-acid, then a water molecule is lost, giving the anhydride (Figure 3) [27].

The biosynthetic origin of necrodaanhydride (Figure 2) is postulated in trans-α-necrodol, a component of the same plant, which is allyl-oxidized to ketoalcohol **I**. Further oxidation of the primary alcohol to carboxylic acid allows easy tautomerization to enol **III**, which undergoes degradation with dioxygenases, following the B1 pathway to **IV**. Finally, decarbonylation and dehydration give rise to the anhydride function of the necrodanhydride [28].

#### 2.2.2. Route B2

Compounds possessing α-hydroxyketone or 1,2-diketone moieties can undergo C-C bond cleavage through alkylperoxo intermediates type **I** (as a result of the addition of peroxoderivatives). The mechanism has been described following the flavin cycle. This cycle, catalyzed by the enzyme peroxiflavin, consists of two half-reactions and the oxidation of α-hydroxyketones and 1,2-diketones, the most representative examples of which are orthoquinones (these coming from oxidation catalyzed by oxidases of catechols) that can lead directly to anhydrides. The corresponding transformation is an O-insertion between the two carbonyls mediated by Baeyer–Villiger oxygenases (Figure 4) [18,29].

The biosynthesis of salviapritin B has been postulated from the abietane miltirone, both natural products having been isolated from the plant *Salvia prionitis* (Figure 3). Miltirone undergoes hydroxylation at C7 and dehydrogenation at the C5–C6 bond. When the hydroxyl group at C7 is removed, the C20 methyl undergoes a Wagner–Meerwein-type migration, giving rise to a carbocation **I** at C10. This highly unstable intermediate gives rise to saprioorthoquinone after cleavage of the C4–C5 bond, which results in the aromatization of the B-ring, and olefin formation at C3–C4 by proton elimination. Saprioorthoquinone, a natural product also found in *S. prionitis*, undergoes O-insertion catalyzed by Baeyer–Villiger oxygenases, giving rise to the cyclic salviapritin B anhydride. This type of anhydride formation is very common in terpenes, especially in diterpenes.

### 2.3. Route Type C

The C-type pathway is a peculiar route in the formation of naturally occurring terpenic cyclic anhydrides. This biosynthetic pathway is characterized by the different oxidation of different types of furan precursors present in different terpenic skeletons (routes C1–C3). These oxidations of furans are carried out by oxidase or oxygenase enzymes which, although following a different chemical evolution, both tend to converge in a final cyclic anhydride.

#### 2.3.1. Route C1

In natural furans, oxygen can be incorporated into the metabolic pathway to anhydrides by oxidases, generating a monoepoxide. The mono-epoxy type intermediates are labile and are transformed into dialdehyde by rearrangement. On the dicarbonyl, the action of monooxygenases leads to the formation of di-acid (Figure 5) [30,31].

This type of formation of terpenic anhydrides occurs in very few cases, one of the few examples being the anhydride triterpenes with a limonoid skeleton present in *Azadirachta indica*. Thus, the biogenesis of meliacinanhydride (limonoid anhydride isolated from *A. indica*) should start from furan derivative I found in the same plant, via the C1 route. According to this proposal, furan **I** undergoes oxidation by monooxygenase (cytochrome P450) giving rise to the corresponding epoxide intermediate, which evolves into its respective *cis*-enedial **II**. This enedial undergoes oxidation at both ketone groups, producing dicarboxylic acid, which after dehydration evolves to the characteristic maleic anhydride fragment of the meliacinanhydride compound (Figure 4) [30].

#### 2.3.2. Route C2

Some enzymes mimic the oxidation of furans with singlet oxygen by means of a concerted Diels–Alder reaction, where oxygen acts as the dienophile and furan the diene (Figure 6). Both species are significantly reactive and give rise to a 6-membered cyclic endoperoxide. This endoperoxide is labile and can easily rearrange to a diepoxyfuran. Diepoxyfurans evolve to anhydrides, having epoxylactones or hydroxylactones as intermediates. Epoxylactones are responsible for the formation of the anhydrides originating from the C2 route (Figure 6) [31,32,33].

In the biosynthesis of anhydrides generated via the C2 route, the addition of molecular oxygen plays a decisive role. The possible formation of two diastereomeric endoperoxides in some skeletons, and their subsequent evolution involving a rearrangement to spiro derivatives, is a key step in the stereochemistry of the resulting anhydrides. Since the rearrangement reaction of endoperoxides to their relative diepoxyfurans and subsequent isomerization to spiro derivatives is stereospecific, it has been postulated as a clear example of the biosynthesis of virgaurene-anhydrides A and B with eremophyllan skeletons. The biosynthesis of both anhydrides starts from the same precursor, which evolves according to the stereochemistry of the endoperoxide formation to two stereochemically different spiranes (Figure 5) [34].

#### 2.3.3. Route C3

This pathway implies the involvement of furano-endoperoxide **I**, which can evolve to hydroxylactone **II** after undergoing homolytic cleavage on the peroxo bond. Intermediate **II** finally evolves to the corresponding anhydride after a Grob fragmentation (Figure 7).

Grob fragmentation is present in the biosynthetic pathway of many natural products. This reaction can be catalyzed by enzymes or occur via unstable intermediates that spontaneously evolve to more stable ones [35,36]. Grob fragmentation requires the presence of a 1,3-hetero-diffunctionalized cluster. One of the functions contains a negatively charged oxygen atom or at least one pair of free electrons and the other a good leaving group. Generally, it usually results in a carbonyl group and an olefin produced by fragmentation between the C1 and C2 atoms. Sometimes the Grob scission leads to a domino reaction, with the formation of new bonds or rearrangements. In terpenes, this type of reaction leads to structural modification, resulting in secoterpenoids and norterpenoids [35,36].

Secovirgaurenol B is an anhydride sesquiterpene isolated from the Chinese species *Ligularia virgaurea*. The biosynthesis of this compound has been postulated to start from sesquiterpenic furan **I**, which is oxidized to endoperoxide **II**. Intermediate **II** evolves to hydroxylactone **III**. This species possesses an appropriate 1,3-difunctionalization and finally undergoes a Grob-type cleavage, leading to C-8-C-9 bond breakage and the formation of the maleic anhydride moiety. This example clearly illustrates the C3 pathway proposed in this review (Figure 6) [37].

The natural products possessing a cyclic anhydride reviewed here are listed in tables ordered by the molecular size of their terpenic, mono-, sesqui-, di-, sester- and triterpene skeleton. In addition, a set of direct derivatives, such as imides, found in natural extracts are collected. In addition, some natural products that were initially identified as anhydrides and whose structure was later revised towards lactone–carboxylic acid are collected. For each anhydride, a route of formation in the final oxidation phase (routes A–C) and, in most cases, an immediate biosynthetic precursor have been assigned. A list of references to their natural origin is included.

## 3. Monoterpenic Anhydrides

### 3.1. Necrodanes

Necrodanhydride (**1**), a monoterpenoid with a nornecrodane skeleton, was isolated from *Lavandula luisieri* (Rozeira) (Table 1). This compound was detected in wild plants collected in the southwest of the Iberian Peninsula and in domestic plants. This natural product is only present in the aerial parts and, in small proportions, in hexane extracts and in essential oil and hydrolates.

Necrodanhydride proved to be a potent nematicide against *Meloydogine javanica*, with an LD_50_ and LD_90_ of 0.24 and 0.52 µg/mL. In addition, it exhibits phytotoxic properties in tests on *Lactuca sativa* and *Lolium perenne* [28] and ixodicidal activity against *Hyalomma lusitanica* [38].

### 3.2. Iridanes

The anhydrides iridoid **2** and *seco*riridoid **3**, possessing an iridane skeleton, have been described (Table 2). Compound **2** was identified in essential oils from *Stachys viticina* [25] and from *Juniperus communis* L. (common juniper). This product was reported to possess antibacterial, cytotoxic and anti-inflammatory activities [39]. Jasminanhydride (**3**) was isolated from the methanolic extract of the medicinal plant *Jasminum grandiflorum* L. [40,41].

The biosynthesis of both **2** and **3** is proposed to follow route A, with chrysomelidial **2p** and dialdehyde **3p** considered to be their precursors. Compounds **2p** and **3p** were found in olive oil and in *J. grandiflorum*, respectively [42,43].

## 4. Sesquiterpenic Anhydrides

### 4.1. Norsesquiterpenoids

Compounds (**4**–**11**) are norsesquiterpenes that have their common biosynthetic precursor in farnesol **4p** [44,45,46,47,48,49,50,51,52,53,54,55,56] (Table 3). The skeleton of these compounds is characterized by possessing the structural grouping bicyclo 3,5-epoxycyclohexane (oxabicycloheptane), in addition to the anhydride functional group, or, in some cases (**8**–**11**), related imides.



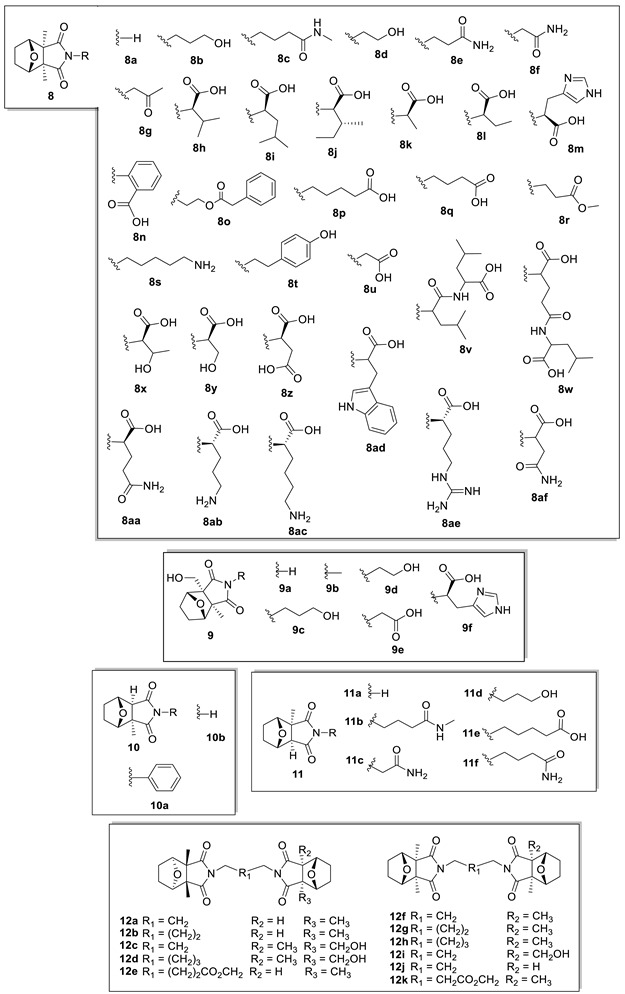



Cantharidin (**4**), the most relevant compound of the group is produced in two insect families, blister beetle (*Coleoptera meloidae*) and false blister beetle (*Coleoptera oedemeridae*) [57,58,59]. In 1810, P. Robiquet isolated cantharidin from, for the first time, *Lytta vesicatoria* (“Spanish fly”) [60], and its structure, proposed by Gadamer and his team in 1914 [61], was finally established in 1951 when Stork synthesized cantharidin stereoselectively [62].

Three anhydrides with a similar skeleton to cantharidin were also described. Two of them, with molecular formulas of C_9_H_10_O_4_, were enantiomers, and were called (+) and (−) palasonin (**6**–**7**). Recently, the existence of 10-hydroxy-cantharidin (**5**), with a molecular formula of C_9_H_10_O_5_, was also reported. Stereoisomers **6** and **7** have biosynthetically lost one of the angular methyls of cantharidin, while in **5**, one of the methyls has been oxidized to alcohol [46,47].

No plants producing cantharidin (**4**) have been described; however, (+) palasonin (**6**) was first isolated from the seeds of the Indian medicinal shrub *Butea frondosa* (*B. monosperma*) Lynn [48]. Palasonin was isolated for the first time in insects such as *Hycleus lunata* (African blister beetles) [54]; it was also found in *Hycleus polymorphus* and *Mylabris quadripunctata* from Southern France [59] and in *Cyaneolytta* sp. African species collected in Nairobi [63], but in a significantly lower proportion than cantharidin. Both (+) and (−) palasonin enantiomers are produced in blister beetles with low enantiomeric excess, with (*R*)-(+) being the major one [47]. By contrast, in *Mylabris cichorii*, only the (−)-palasonin enantiomer has been found.

More than 40 imides derived from cantharidin **4** and its alcohol **5**, formed by the reaction of the anhydride function with an amine, sometimes a known amino acid, are known. Some of them were given proper names, such as cantharimides A–J (**8a**–**8g** and **9b**–**9d**), canthaminomides A–F (**8h**–**8m**), cantharylmides A–B (**8n**–**8o**), cantharacidines A and C (**8p**–**8q**), cichormides A–D (**9e**–**9f** and **8r**–**8s**) and others. Cantharidinimide (cantharimide A) was found in *Lytta vesicatoria* and *Mylabris mongolica* by alkaline extractions of these insects [64]. *Mylabris impressa* was the first of its genus to be reported to possess cantharidin, along with palasonin and cantharidinimide (**8a**) [65]. Along with cantharidin and palasonin, cantharidinimide and palasoninimide have been found in two species of blister beetles of the genus Hycleus, that is, *H. scabiosae* and *H. lunata*. [66]. Noteworthily, large quantities of **8**–**9** have been discovered in the blister beetle *Mylabris cichorii* L. and *M. phalerate* [46,49,50,51,52,53].

Several palasonin-derived imides were also reported, including palasimide (**10a**), isolated from the pods of the medicinal plant *Butea monosperma* with a yield of 0.0008% [55], and palasoninimides A–D (**11a**–**11d**) and cantharacidines B and D (**11e**–**11f**), also from *M. phalerata* [49,53]. Additionally, 11 dimers, consisting of two derived cantharimide monomers linked by a hydrocarbon chain, were also identified in *M. phalerata*, including the so-called dicantharimides A–E (**12a**–**12e**) [50,51,53,56].

The multiple applications and biological activities of both these molecules and the insects that contain them have been described. Thus, cantharidin (**4**) acts as a chemical defense of insect producers against predatory organisms and has been used since ancient times for its biological activities [67]. In this regard, *Mylabris phalerata* and *M. cichorii*, both containing **4**, are the first insects historically used in traditional medicine since ancient times for the treatment of innumerable ailments such as boils and hemorrhoids, deep root ulcers, poisonous worms, tuberculous lymphadenitis fistulas, the elimination of dead tissues and scrofula, as abortifacients, purgatives, anti-rabies, to treat neoplasms, lice and nits, for vermicide and as an antiparasitic treatment in livestock [68,69]. *Lytta vesicatoria* was used due to its high cantharidin content (<1%) in treatments throughout the Middle Ages as a verrugicide, vermicide, and anti-acne. “Spanish fly” also had a popularized use as a sexual stimulant in the early 17th century but proved toxic to the urinary system and lethal at mg/kg concentrations. Many of these applications have been passed down from generation to generation as home remedies and are still used today [68].

In medicine, cantharidin (**4**), palasonins (**6** and **7**) and their imide derivatives **8**, **10** and **11** were found to be active against cancerous lines in humans. There has been antitumor activity of cantharidin against skin cancer A431, bladder cancer T24, RT4, non-small-cell lung cancer (NSCLC) NCI-H460, A549, H358, colorectal cancer COLO 205, hepatocellular carcinoma HepG2, Hep3B, gastric cancer BGC823, MGC80, cholangiocarcinoma QBC939, breast cancer cell MCF-7, pancreatic cancer PANC-1, CFPAC-1, oral carcinoma SAS, SCC-4, CAL-27, TCA8113, UMSSC, ovarian cancer HO-8910PM, OVCAR-3, prostate cancer PC-3, DU-145, breast cancer BT-549, T-47-D, MDA-MB-435 and CNS cancer U251 and SNB-19, among others [70]. A compilation that includes the cantharidin and norcantharidin IC_50_ values for more than 50 human cancer cell lines representing twelve different types of human cancers has been recently published [71]. Moreover, the activity of cantharidin against kidney cancer has also been reported, including TK-10, ACHN, RXF393, SN-12C, CAKI-1, 786-0, UO-31 and A498, with IC_50_ values ranging from 3 to 20 μM. It has been widely accepted that the anticancer activity of cantharidin is mainly due to the inhibition of PP2A (protein phosphatase type 2A) and HSF-1 (Heat Shock Transcription Factor 1). In this regard, cantharidin can affect various cell signaling pathways, but MAPK, Bcl2/Bax, JNK, NF-κB, ERK, PKC, β-catenin, Wnt/β-catenin, PI3/AKT and PIk3/ATK/mTOR are recognized as its potential molecular targets. Other derivatives such as norcantharidin were also reported to inhibit the Wnt signal pathway via promoter demethylation of Wnt inhibitory factor-1 in human non-small-cell lung cancer [72].

Additionally, these compounds are mainly used in dermatology for their vesicant properties in the removal of tattoos, for the treatment of plantar warts in podiatry and the treatment of the virus *Molluscus contagiousum*. In this regard, cantharidin at concentrations of 0.7% is effective in mitigating the adverse effects of the virus, including the appearance of blisters [69]. Different authors point out the structural importance of cantharidin, especially its cyclic anhydride, as being responsible for the bioactivity of these compounds [73,74,75].

In veterinary tests, the ethanolic extract containing **6** proved to be anthelmintic, whereas **7** showed good in vitro activity against the human intestinal parasitic nematode *Ascaris lumbricoides*, proving to be the active principle of this plant [76].

Cantharidin (**4**) also possesses sexual and ecological activities. It has been shown that adult male blister beetles produce much more cantharidin than females [59] Specifically, although both sexes synthesize cantharidin in their larval stage, females hardly produce cantharidin in the adult stage, so their cantharidin level comes from their partners during mating [77]. This demonstrates the utilization of **4** as a sexual attractant for females. It also acts as a powerful attractant for several groups of insects known as canthariphilous [78,79] and, more unusually, for other animals such as the *Otis tarda* bird [45]. These animals have developed a tolerance to these toxic compounds and may ingest them and benefit from their biological activity or use them as a chemical signal in the search and location of food. This introduction into the food chain is the reason why cantharidin, which is only produced by blister beetles and false blister beetles, has been detected in a great diversity of organisms [45,78,79]. Extracts of common oil plant *Berberomeloe majalis* (blister beetles) containing **4** were toxic against one protozoan (*Trichomonas vaginalis*), one nematode (*Meloidogyne javanica*), two insects (*Myzus persicae* and *Rhopalosiphum padi*) and one tick (*Hyalomma lusitanicum*). This supports the anti-parasitic hypothesis for the consumption of that insect by *Otis tarda* [45]. Another very important application of **4** is in pest control. Thus, this anhydride is used in insect traps because of its capacity to attract a great variety of canthariphilious insects responsible for numerous pests causing important damage in agriculture [80].

Regarding the imide derivatives of these anhydrides, cantharimides A, D (**8a**, **8d**), **8t** and **12g** were found to be potent inhibitors of the HBV virus, with IC_50_ values of 62, 42, 58 and 19 μM [49].

Canthaminomide F (**8m**) showed weak cytotoxicity on three cancer cell lines (HepG2, A549 and MDA-MB-231 cells), with IC_50_ values of 15.70 ± 3.06, 20.67 ± 4.05 and 23.79 ± 4.54 μM, respectively [56].

Cichormides B and D (**9f**,**8s**) and (−)-palasonin (**7**) were able to inhibit renal fibrosis in vitro at 40 µM by inhibiting the expression of fibronectin and collagen I expression in TGF-β1-induced NRK-52e cells [46].

Considering the biosynthesis of cantharidin (**4**), it was initially hypothesized that either this compound is generated from the acetate pathway, or it derived from a monoterpene produced after the unusual head-to-head coupling of two isoprene units [81]. However, isotopic labeling experiments determined that its precursor was the sesquiterpene farnesol [82,83,84]. Particularly, isotopic labeling showed that farnesol is incorporated into cantharidin in *Litta vesicatoria* and degraded by C4–C5 and C7–C8 bonds [85]. In the same sense, it has been shown that methyl farnesoate is incorporated into cantharidin [77]. Du et al. studied several genes related to cantharidin biosynthesis in *Mylabris cichorii* [86] and postulated that its biosynthesis proceeded most likely through the MVA pathway, rather than the MEP pathway or a mixture of both pathways. A differentially expressed genes (DEGs) genetic study of cantharidin biosynthesis in the insects *Hycleus cichorii* and *Hycleus phaleratus* [87] found 19 cantharidin biosynthetic pathway genes belonging to the MVA pathway. Of these, farnesol dehydrogenase is the one expressing the highest number of copies. Other relevant ones were hydroxymethylglutaryl-CoA reductase, hydroxymethylglutaryl-CoA synthase, isopentenyl difosfate isomerase and acetylcoenzime A c-acetyltransferase. In general, higher amounts were expressed in females than in males. In relation to the previous study, three genes involved in cantharidin biosynthesis were detected in *Epicauta chinensis*, namely, methyl farnesoate epoxidase, juvenile hormone acid O-methyltransferase and juvenile hormone epoxide hydrolase [88].

Using the insect *Epicauta tibialis*, RNA expressions in adult males and females were compared by a quantitative real-time PCR (qPCR). Seven regulated DEGs were found in adult males—including acetyl-CoA C-acetyltransferase, hydroxymethylglutaryl-CoA synthase, hydroxymethylglutaryl-CoA reductase, isopentenyldiphosphate delta-isomerase, NADP+-dependent farnesol dehydrogenase, farnesyl diphosphate phosphatase and aldehyde dehydrogenase—that were potentially involved in cantharidin biosynthesis [83,84,85,89].

Once it is accepted that farnesol (**4p**) is the biosynthetic precursor of cantharidin (**4**) in the process, these sesquiterpenes necessarily must be degraded by losing five carbons, C1, C4–C7 and C14. Of the five, four have been shown to be lost by degradation of the C4–C5 and C7–C8 bonds [82]. It is well documented that oxygenases from insect juvenile hormone synthesis, which are present in cantharidin-producing insects, are involved in the oxygenation of C-H bonds and olefin epoxidation in farnesol [83,84,85,86,87,88,89]. It is relevant to mention that between the precursor farnesol (C15) and cantharidin (C10), no biosynthetic intermediates have ever been detected in cantharidin-producing organisms, although it is proposed that the biosynthesis goes through different juvenile insect hormones and derivatives [88].

These results seem to indicate that the reactions starting from a polyoxygenated farnesol intermediate, such as subsequent oxidations, degradation and the construction of the oxabicyclocyclohexane system, must be attentively efficient and rapid. In this regard, it has been postulated that the biosynthetic precursor of palasonin (**6** and **7**) is cantharidin itself, after degradative oxidation of the methyl groups. The detection of cantharidin hydroxyderivatives on one of the methyls in blister beetles (*M. chicorii*) supports this hypothesis [46].

Upon this background knowledge, we postulate here a hypothetical biosynthetic route of cantharidin (**4**) from farnesol **4p**, trying to shed some light on the degradative steps of the biosynthesis of this anhydride (Figure 7). Three well-differentiated phases are proposed. In the first one, farnesol is functionalized to intermediate **II**, which has a suitable functionality to allow the construction of the key C4–C8 bond in the process. This phase involves enzymes known to be present in insects that biosynthesize cantharidine, in particular, those belonging to the insect juvenile hormone formation pathway (C–H bond oxygenases, epoxides, etc.) [84,88]. We hypothesize that the formation of both the double bond at C8–C9 and the allylic alcohol at C2–C4 are efficiently achieved via epoxides at C2–C3 and C6–C7, as in **I**. Thus, the opening of the C2–C3 epoxide with the elimination of H4 (hydrolysis to diol and dehydration of the tertiary hydroxyl could also be possible) and a similar process on the C3–C6 epoxide, with the elimination of alcohol at C5, leads to intermediate **II**. Additionally, the C12 methyl has also been oxidized to carboxylic acid.

In the second phase, the cyclohexane present in cantharidin possessing the C4–C8 bond will be originated. A highly efficient reaction towards that goal would be an intramolecular Diels–Alder reaction catalyzed by a Diels–Alderase, an enzyme type known to act in different organisms [90,91]. In this case, the process is favored by the inverse electronic demand and the configuration of intermediate **II**, which allows this electronic interaction without impediment. Moreover, this type of process should allow a high stereospecific control in the formation of the bicyclic intermediate **III**. Obviously, this proposal is merely speculative, and this phase of the biosynthesis could take place by more common pathways such as cyclase-catalyzed cyclization. This bicyclization is a key step, since it involves the formation of the C4–C8 bond present in cantharidine prior to the subsequent degradation of the four carbons C5, C6, C7 and C14.

The third phase involves degradation processes leading to the required loss of five carbon atoms. Intermediate **III** already contains the complete carbon skeleton of cantharidine, and its degradation should not imply the formation of two separate fragments, since these would necessarily have to be condensed again. The presence of an electronically rich trisubstituted C6–C7 double bond on the cyclopentane is prone to undergo dioxygenase-mediated degradation to give **IV**. This intermediate has a 1,5-dicarbonyl grouping that directly loses two carbon atoms, via a RetroMichael process, to afford **V**. At this point, two additional carbons are lost via hydration and a retro Claisen transformation to give **VI**. This degradative model offers remarkable efficiency, since only three steps build up the 4,8 bond and degrade four carbon atoms. From **VI**, the action of reductase on the C4 carbonyl gives the corresponding alcohol, whereas a cyclase originates the oxygen bridge. Finally, with a dioxygenase-mediated degradation, the diol produces **VII**, which by dehydration leads to cantharidrin **4**.

### 4.2. Germacranes

In 1984, Goren et al. isolated, from *Smirnium rotundifolium*, *seco*germacran **13**, whose biosynthetic origin must be in furanodiene **13p** [92] (Table 4). The reaction of **13p** with singlet oxygen leads to endoperoxide **I**, which after the reductive opening of peroxo and oxidation at C13 produces the intermediate hydroxylactone **II**. For the last step involving the degradation of the germacrane skeleton, a pericyclic reaction is postulated, where -OH at C10 is protonate, with a concomitant carbonyl formation (Figure 8). The same anhydride was found in the marine gorgonian coral organism *Menella* sp. (Plexauridae) at a depth of 100 m [93].

### 4.3. Drimanes

**Table 5 biomolecules-14-00955-t005:** Drimane anhydrides.

Compound	Precursor and Route	Source	Reference
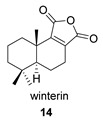	Route A 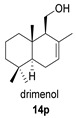	*Drimys winteri* *D. confertifolia* *Polygonum hydropiper* *Dendrodoris carbunculosa*	[94,95,96,97]
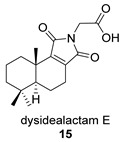	Route A 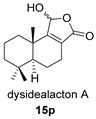	*Dysidea* sp.	[98]
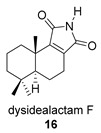	Route A 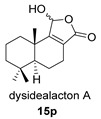	*Dysidea* sp.	[98]
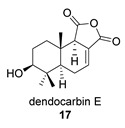	Route A 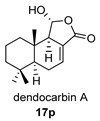	*Dendrodoris carbunculosa*	[92]
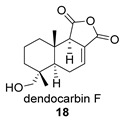	Route A 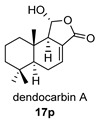	*Dendrodoris carbunculosa*	[95]
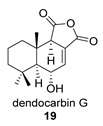	Route A 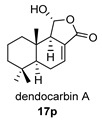	*Dendrodoris carbunculosa*	[95]
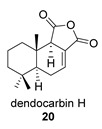	Route A 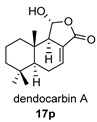	*Dendrodoris carbunculosa*	[95]

The drimane anhydride (+) winterin (**14**) was first isolated from the wood of *Drimys winteri* and *D. confertifolia* [96] (Table 5). Its full spectroscopic description was reported in 2011 [97].

Khushi et al. isolated glycinyl imides **15** and **16** (dysidealactam E and F) from marine sponges of the Great Barrier Reef: These compounds proved to be active in carcinoma cytotoxicity tests at a 30 μM concentration [98].

Four anhydride drimanes (dendocarbins E–H) (**17**–**20**) were found in the Japanese nudibranch *Dendrodoris carbunculosa*. They showed moderate cytotoxic activity, with an IC_50_ (9–10 μg/mL) against several tumor cell lines [95].

### 4.4. 1,3-Cycloeudesmanes

The terpenic anhydride lindenanolide E (**21**) with a *seco*-eudesmane skeleton was isolated from the roots of *Lindera chunii* (Lauraceae) (Table 6). Its biosynthetic precursor is considered to be the furan derivative lindeneol **I**, also existing in the same plant [99]. The biosynthesis is hypothesized to belong to route C, where the furan undergoes singlet oxygen-type oxidation to the corresponding cyclic peroxide **II**, which isomerizes to 7,8;12,13-diepoxide **III**. The electrophilic opening of the 7,8-epoxide causes a rearrangement from C9 to C7, giving rise to spirointermediate **IV**. A new opening, now of 12,13-epoxy via Grob fragmentation, leads to *seco*-eudesmane **V**. Oxidation of this intermediate via monoxygenase and isomerization of the double bond leads to anhydride **21** (Figure 9). The biosynthesis of lindenanolide E (**21**) is a striking case, since the formation of the anhydride occurs via the C2 and C3 pathways.

Chloranthalactone A (**22p**) is an abundant lindenane-type sesquiterpenoid found in plants of the Chloranthaceae family. This substance is reported to easily suffer different addition reactions to generate complicated lindenane conjugates [100]. Most of these molecules possess important bioactivities, including cytotoxic, anti-HIV, antiinflammatory and antimalarial effects. A phytochemical study of the leaves of *Sarcandra glabra* led to the isolation of the anydrides sarglalactones D and E, among other 8,9-*seco*lindenane-type sesquiterpenoids [100]. These dimers significantly reversed the multidrug resistance of MCF-7/doxorubicin (DOX) cells and increased the sensitivity of U2 OS cells to DOX by downregulating HMGB1 expression.

*Chloranthus japonicus* is widely distributed in East Asia, including China, where it is used in folklore medicines for the treatment of traumatic injury, rheumatic arthralgia, fracture, pulmonary tuberculosis, and neurasthenia. A new dimeric lindenane sesquiterpenoid, named chlojapolactone B (**23**), was isolated from this plant [101]. Chlojapolactone B showed moderate inhibition against TNF-α, with an IC_50_ value of 76.16 μM.

A third seco-lindenane disesquiterpenoid anhydride, chloramultiol G (**24**), was isolated from *Chloranthus multistachys* [101], whose root tissue has been used in Chinese folk medicine to treat bone fractures, lumbocrural pain and pruritus. Route C3 has been proposed to generate this compound from the corresponding precursor (**24p**). A key step in the process is an endo Diels–Alder reaction between two derivatives of chloranthalactone A and a Grob fragmentation of the thus-originated dimer (Figure 10).

### 4.5. Guaianes

In the medicinal plant *Chloranthus henryi*, *seco*guaiane anhydride **25** has been isolated [102] (Table 7). This compound must be formed biosynthetically from a furanic precursor, involving the addition of ^1^O_2_, isomerization and Grob fragmentation. However, this precursor has not been found in the plant.

### 4.6. Eremophilanes

*Ligularia virgaurea* is the most widespread native grass in the alpine meadow grasslands of the Qinghai–Tibet Plateau, China, and is an eremophilane-producing species with a high intraspecific diversity of substituents. From its roots, anhydride **26** with a *seco*-eremophilane skeleton was identified, as well as its precursor, furan **26p** [36] (Table 8).

### 4.7. Tremulanes

Coriolopsin A (**27**) was identified in the ethanol extract of the mycelium and solid medium of the endophytic fungus *Coriolopsis* sp. J5, growing in the healthy branches of the plant *Ceriops tagal* [103] (Table 9).

From the pathogenic wood-rotting fungus *Irpex lacteus*, used in traditional Chinese medicine as an antibacterial, anti-inflammatory and diuretic, several sesquiterpenes with a tremulane skeleton were isolated and identified. Three of them, irlactam A (**28**) and irpexolactins A and B (**29** and **30**), are lactams that originated after the reaction of the corresponding anhydrides with aminoethanol [104,105]. All these four compounds were biosynthesized following the A route.

### 4.8. Bakkanes

In the plant *Ligularia virgaurea* (Maxim.) collected in China, two additional acid anhydrides, virgaurene-anhydride A (**31**) and virgaurene-anhydride B (**32**), containing a bakkane spirosesquiterpene skeleton, have been found [34,106] (Table 10).

### 4.9. Aristolanes

In 2017, Cheng et al. described the isolation of aristolanhydride **33** from the roots and rhizomes of *Nardostachys chinensis* [107]. However, in 2019, the structure was corrected to the corresponding dicarboxylic acid **33c** by Wang et al. [108] (Table 11).

Previously, in 2012, *nor*-aristolane **34** was isolated from the same plant [109]. Its biosynthesis is hypothesized to proceed from aristolane **34p**, which evolves to species **III** via compounds **I** and **II**, also present in the plant. From **III** a B route cleavage is proposed to produce **IV**, which is decarboxylated and subsequently oxidizes to give **V**. Finally, after a series of oxidations and dehydration, anhydride **34** is generated (Figure 11).

### 4.10. Thujopsanes

The Asian medicinal plant *Cissus quadrangularis* is well known for its utilization in the treatment of skin contaminations, blockage, piles, lack of health, weight loss, ant-inflammatory and anti-ulcer treatments and bone crack healing. From this species, the anhydride northujopsane (**35**) was extracted [110] (Table 12). The proposed precursor of **35** could be mayurone (**35p**), following the B1 pathway. Enone **35p** was described in *Thujopsis dolabrata* [111].

### 4.11. Pseudo Sesquiterpenes

The pseudo sesquiterpene called paniculoid or paniculatic anhydride (**36**) was isolated from the seeds of the medicinal plant *Koelreuteria paniculata*, commonly called China soapwort because of its ancient cleaning use, which is attributed to its content of saponins [112,113] (Table 13). Anhydride **36** possesses a novel skeleton, and no possible precursor has been found in the plant. In any case, its classification as a terpene can be questioned from our point of view.

## 5. Diterpenoids

### 5.1. Labdanes

Compound avxanthin B (**37**) was isolated from the rhizomes of *Amomum villosum* (Table 14). It exhibits significant activity as anti-inflammatory, with an IC_50_ value of 11.0 μM, inhibiting nitric oxide production. It also showed an interesting inhibitory activity on α-glucosidase, with a ratio greater than 90% at IC_50_ 21.1 μM [114].

From the endophytic fungus *Hypoxylon* sp. from the plant *Bruguiera gymnorrhiza*, the anhydride hypoxyterpoid B (**38**) was extracted. Route B is proposed for its biosynthesis from **38p**, also present in the plant [115].

### 5.2. Cassanes

From the roots of *Atalantia buxifolia*, the patented compound **39** was isolated (Table 15). This substance proved to possess a strong cytotoxicity and anticancer effect in several breast cancer cell lines [116]. Although a suitable precursor has not been identified in *A. buxifolia*, its biosynthetic origin can be postulated from a furano-cassane such as caesalmin E_2_ (**39p**), present in *Caesalpinia minax* [117]. The pathway is proposed to comprise an oxidation process with singlet oxygen, followed by isomerization, rearrangement of the epoxide intermediate and Grob-type fragmentation (Figure 12).

Diterpenic anhydride **40** was isolated from the dried roots of *Ranunculus ternatus*. It has been patented as a neuroprotective drug [118].

### 5.3. Abietanes

Abietanes have been isolated mainly from plants of the Labiatae family. This family contains a good number of medicinal plant species that originate a great diversity of natural products with a different biosynthetic origin, diterpenoids being one of the most interesting ones. The anhydride terpenoids reviewed here reach the final anhydride grouping by a process that is common to practically all of them, the degradation of O-quinones following the B route (Table 16).

Obtuanhydride (**41**) was first isolated in 1998 from *Chamaecyparis obtusa* var. *formosana* Rehd. (Taiwan Hinoki; Cupressaceae). This species is an important building material in formosa because of its strong resistance to wood-decaying fungi [119]. Obtuanhydride isolated from the lichen *Cladonia rangiferina* was tested as an antibacterial against *Staphylococcus aureus* (MRSA), showing moderate activity [120].

Sanguinone A and its 17-hydroxy derivative (**42** and **43**) are glandular pigments isolated from the plant *Plectranthus sanguineus* Britten [121] collected in Milawi. Other quinones and hydroquinones were also isolated from this species, including the precursors of these anhydrides (**42p** and **43p**).

Normiltirone (**44**) was isolated in 2019 from the roots of *Salvia miltiorrhiza* [122]. These roots have been used in Chinese folk medicine as a sedative and tranquilizer, including the treatment of coronary heart disease. Its probable precursor, miltirone (**44p**), was also found in the same plant and has a wide range of biological activities [123].

Cryptomanhydride (**45**) was isolated from the leaves of the Japanese cedar *Cryptomeria japonica*, and is characterized by a unique skeleton consisting of an abietane diterpene and a *p*-cymene monoterpene [124]. A possible biosynthesis of this compound would involve the a-alkylation of 6-ketosugiol (**45p**), a ketone also present in *C. japonica*, with the corresponding *p*-cymene derivative. Subsequent oxidation of the C-ring at position 11 would enable the generation of the anhydride ring (Figure 13).

Taxodinoid A (**46**) is an anhydride with a bis-abietane structure, isolated from the cypress *Taxodium distichum*, widely distributed in China, and used in folk medicine [125]. Along with it, three other dimeric structures were isolated, including **46p**, a substance that can be proposed as its precursor. The effect of the metastatic activity of these compounds has been studied, so that it has been found that **46** suppresses the metastasis of U2 OS tumor cells in a dose-dependent manner.

*Salvia prioritis* is used in Chinese folk medicine for the treatment of tonsillitis, pharyngitis and bacillary dysentery. Notable among its constituents are diterpenes with an abietane structure, some of them being dimer structures such as compound **47** [126]. The constituting monomer unit of **47** must correspond to arucadiol (**47p**), which has also been identified in *S. prionitris* [127].

Again from *S. miltiorrhiza*, a series of anhydrides known as tanshinane anhydrides have been reported (**48**–**51**) [128,129]. The precursors of these compounds are the quinones **48p**–**51p**, also present in extracts of this plant. Interestingly, interesting biological properties of these anhydrides have been reported; thus, dihydrotanshinone anhydride was reported to possess significant anti-inflammatory activity (an inhibitor of tumor necrosis factor-α, interleukin-1β and interleukin-8 production in THP-1 macrophages) [128]. Additionally, all four of these anhydrides were proven to be highly potent inhibitors (Ki < 1 nM) and can inactivate human cholinesterases both in vitro and in cell culture systems (tanshinone anhydrides react with hCE1 active-site serine, forming covalent complexes) and can modulate the metabolism of the esterified drug oseltamivir [129].

Closely related structurally to these tanshinane anhydrides, the presence of the 1,2-hydroderivative salmialbanone (**52**) in *S. miltiorrhiza* has been reported [130].

Castanol A (**53**), an hydroxy derivative of compound **49**, was isolated from *S. castanea* and *S. grandifolia* [131,132], both widely distributed in China. Its precursor must be hydroxytanshinone IIA (**53p**), a natural compound present in other species of *Salvia* [133].

Hydroxyanhydrides (**54**–**56**) are the product of the biotransformation of cryptotanshinane (**54p**) by the fungus *Mucor rouxii* AS 3.3447. Its inhibition rate against influenza A virus and its cytotoxicity were tested at 10 μM, obtaining values of inhibition higher than 90% and even reaching 97%. It has been postulated that the presence of anhydride and hydroxyl functional groups in the corresponding natural product may contribute to its increased antiviral activities and decreased cytotoxicity [134].

Saruul et al. isolated in 2015 in the dried roots of another lamiaceae, *Caryopteris mongolica*, a series of *ent*-abietanes with a methylcyclopropane moiety [135]. Among them, the caryopteron B–D anhydrides (**57**–**59**) were reported. The biosynthetic origin of these anhydrides must follow the B route, since α-dicetone **57p** is present in the same plant.

The structures of caryopterones B, C and D were revised and corrected to the corresponding carboxylic acids (**57c**–**59c**) on the basis of computational calculations [136].

*Hance* (Labiatae) is a Chinese medicinal plant used as an antiphlogistic, antibacterial and antituberculosis drug. Two compounds with an anhydride structure, namely, salviapritin B (**60**) and saprionide (**61**), have been isolated from the roots of *Salvia prionitis* [137,138]. The biosynthetic precursor to both molecules is postulated to be a 4,5 secoabietane, saprioorthoquinone (**60p**), also present in the same plant. This precursor evolves via intradicarbonyl O-insertion to salviapritin B, through the B route. The biosynthetic route of saprionide should proceed by cyclization of the double bond on the C11 carbonyl, leading to saprirearine (**61p**). A degradative process would lead to salviapritin A, which, by loss of 1 C, would lead to saprionide, following route B (Figure 14) [137,138]. Both saprirearine and salviapritin A have also been isolated from *S. prionitis*.
biomolecules-14-00955-t016_Table 16Table 16Abietane anhydrides.CompoundPrecursor and RouteSourceReference
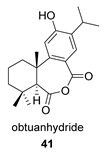
Route B 
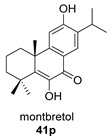
*Chamaecyparis obtusa* var. *formosana*
*Cladonia rangiferina*[119,120]
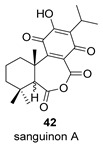
Route B 
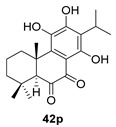
*Plectranthus sanguineus*[121]
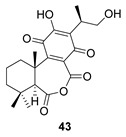
Route B 
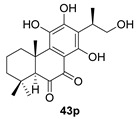
*Plectranthus sanguineus*[121]
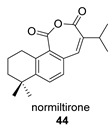
Route B 
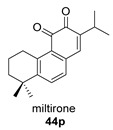
*Salvia miltiorrhiza*[122,123]
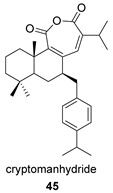
Route B 
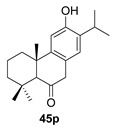
*Cryptomeria japónica*[124]
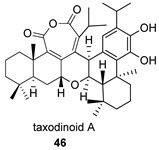
Route B 
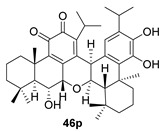
*Taxodium distichum*[125]
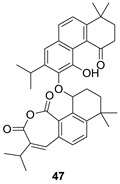
Route B 
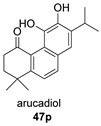
*Salvia prionitis*[126,127]
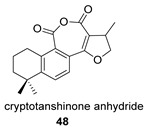
Route B 
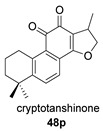
*Salvia miltiorrhiza*[128,129]
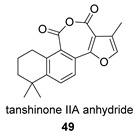
Route B 
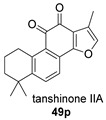
*Salvia miltiorrhiza*[128,129]
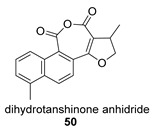
Route B 
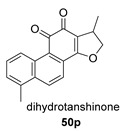
*Salvia miltiorrhiza*[128,129]
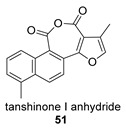
Route B 
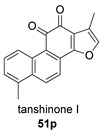
*Salvia miltiorrhiza*[128,129]
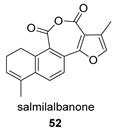

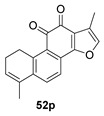
*Salvia miltiorrhiza*[130]
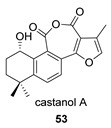
Route B 
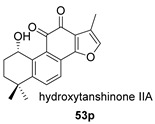
*Salvia castanea**Salvia grandifolia*[131,132,133]
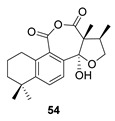
Route B 
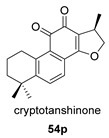
*Mucor rouxii*[134]
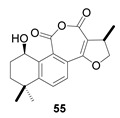
Route B 
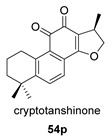
*Mucor rouxii*[134]
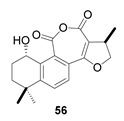
Route B 
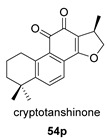
*Mucor rouxii*[134]
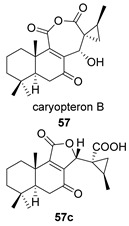
Route B 
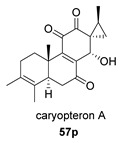
*Caryopteris mongolica*[135]
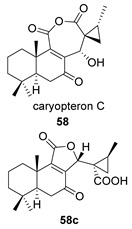
Route B 
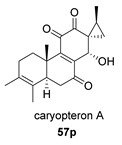
*Caryopteris mongolica*[135]
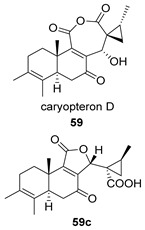
Route B 
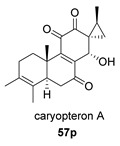
*Caryopteris mongolica*[135]
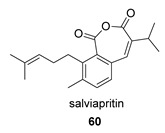
Route B 
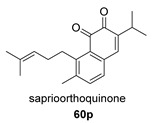
*Salvia prionitis*[137]
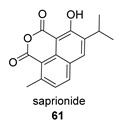
Route B 
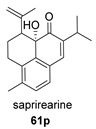
*Salvia prionitis*[138]

### 5.4. Pimaranes

The norpimarane anhydride **62** was isolated from the marine sponge-derived fungus *Aspergillus* sp. Its biosynthetic precursor **62p** was found in the fungus [139] (Table 17).

### 5.5. Fusicozanes

In the agrimony of *Plagiochila sciophila*, the fusicoccane-type diterpenoids fusicosciophin C (**63**) and its precursor fusicosciophin B (**63p**), which should evolve into the related anhydride route B, were isolated (Table 18). Fusicosciophin C exhibited moderate lettuce seed dormancy-breaking activity at concentrations between 1.5 and 37.5 μM [140].

Alterbrassinoid D (**64**), an unprecedent fusicoccane-derived homodimer anhydridide presenting a C-12–C-18′ linkage, was isolated from modified cultures of *Alternaria brassicicola* [141]. The cytotoxicity of alterbrassinoid D against five tumor cell lines (including OCVAR, MDAMB-231, HeLa, HT-29 and Hep3B) was tested. The results proved that this anhydride showed moderate activity against these tumor cells, except for the Hep3B cell, with IC_50_ values ranging from 10.79 to 39.81 μM. Its plausible biogenetic precursor (**64p**) was also found in the same natural source.

### 5.6. Ent-Kauranes

Fujenal (**65**) and fujenoic acid (**66**) were first described as a constituent of the fungus *Gibberella fujikuroi* [142] (Table 19).

The production of fujenal from *ent*-kaurenoic acid has been demonstrated by feeding experiments (Figure 15) [143]. Gibelactol (**65p**), also isolated from *G. fujikuroi* [144], must be an intermediate of the process, which falls under the so-called route A. The plausible precursor of fujenoic acid (**66p**) was likewise described by one of the authors in a mutant of *G. fujikuroi* [145]. Fujenal has been described to inhibit clover infection with *Rhizobium trifolii* [146].

### 5.7. Spongianes

Isolated in 1990 in *Dictyodendrilla cavernosa*, natural product **67** was the first anhydride isolated in a marine sponge [147] (Table 20). Its biosynthetic precursor should be the related furanospongiane **67p**, isolated in other sponges [148], which should give rise to **67**, following the C pathway.

### 5.8. Atisanes

The pentacycloditerpene ahydride cardionidine (**68**) is the only example of an alkaloid with an atisane carbon skeleton. It was isolated from the plant *Delphinium cardiovphetalum* [149] (Table 21). Cardionidine (**68**) must have originated from miyaconinone (**68p**), a diketone alkaloid isolated from *Aconitum miyabei* [150].

### 5.9. Phenanthrenone Diterpenoids

The medicinal plant *Strophioblachia fimbricalyx* has been popularly used to treat cancer, migraine and fevers. From its roots, fimbricalyxanhydride A (**69**) and B (**70**) are extracted, and from its stem and leaves, B (**70**) and C (**71**) [151,152] (Table 22). Fimbricalyxanhydride A was tested for cytotoxicity (KB, MCF7 andNCI-H187 cancer cells) and antiplasmodial activity (*Plasmodium falciparum*, K1 multidrug-resistant strain), showing cytotoxic effects towards NCI-H187 cancer cells and antiplasmodial activity, with IC_50_ values of 5.7 and 3.9 μM, respectively. Fimbricalyxanhydride B and C were tested against A549, HeLa and HepG2 cancer lines, but only **70** showed some cytotoxic effect [151,152].

*Trigonostemon lii* is a shrub or small tree endemic to Southern China, and from the ethanolic extract of its roots and stems was extracted thrigonosomone A (**72**) [153]. This compound contains an extra oxygenated function in its seven-membered cyclic anhydride moiety. Its proposed precursor **72p** is also present in the same plant.

The structures of fimbricalyxanhydride B (**70**) and thrigonosomone A (**72**) were corrected to those represented by **70c** and **72c** [136].

### 5.10. Neoclerodanes

*Salvia divinorum*, also called “magic mint” due to its hallucinogenic effects, has been traditionally used by the Mazatec Indians of Northeastern Oaxaca in spiritual practices. Salvinorin A (**73p**), a non-nitrogenous neoclerodane characterized as a potent and selective kappa opioid receptor agonist, was extracted from this plant (Table 23). From the vaporized smoke of **73p** (whose inhalation produces intense hallucinogenic effect in humans), eight neoclerodane derivatives have been identified, among them, the anhydride epimers couple **73** [154].

## 6. Sesterterpenoids

### Acyclic Sesterterpenoids

Hippolide D **74**, a sesterpenoid with an anhydride-type structure, and two others with a succinimide structure, hippolides A and B **75**–**76**, were isolated from the sponge *Hippospongia lachne* (Table 24). Hippolide A presented cytotoxic activities, with IC_50_ values of 5.22 × 10^−2^, 4.80 × 10^−2^ and 9.78 µM against A549, Hela and HCT-116 human cancers. It also showed moderate PTP1B inhibitory activity, with an IC_50_ of 23.81 µM. The activity of hippolide B (**76**) was moderate, with higher IC_50_ values against HCT-116 and PTP1B [155]. It has been hypothesized that the non-functionalized furane **74p**, which is found in other *Hippospongia* sp. [156], is the precursor of hyppolides D, A and B, which evolve to their final products via route C1.

Compound **77** was isolated from the sponge *Sarcotragus* sp. collected from Korean waters. The compound showed moderate inhibitory activity against isocitrate lyase (ICL), an enzyme that plays a key role in fungal metabolism derived from *Candida albicans* [157]. The precursor is the sesterpenoid **77p**, which evolves to **77** via the C1 route after losing four carbon atoms.

## 7. Triterpenoids

### 7.1. Limonoids

Five triterpene anhydrides with a limonoid skeleton (**78**–**82**) were isolated from different parts of the plant *Azadirachta indica* [158,159,160] (Table 25). It is postulated that the maleic anhydride ring originates in all cases from the corresponding furan, following the C1 route. Tetracyclic triterpenoids **78** and **79** were isolated when the plant was collected in Venezuela [158], whereas meliacinanhydride (**80**) was isolated when it was collected in Pakistan [159]. Limbocidin (**81**) and limbocidin (**82**) were found in the seeds of the plant [160].

The cytotoxicity of compounds **78** and **79** was evaluated in cancer cell lines using HeLa (human cervical epithelioid carcinoma) and PC-3 (human prostate adenocarcinoma). Only **79** showed significant growth inhibition and a low cytotoxicity [158].

Meliacinanhydride (**80**) proved to be of interest as an antiviral, as a result of computational calculations of molecular binding with the hepatitis C virus NS3/4A protease [161]. It turned out that the bioactive compounds presented in their structure both a cyclic anhydride and an unsaturation at C-1.

### 7.2. Lanostanes

Applanhydride A (**83**) and B (**84**) are triterpenes with a lanostane tetracyclic skeleton, found in the fruiting bodies of the mushroom *Ganoderma applanatum* collected in China (Table 26). These terpenes were the first examples of triterpenoids with a 7-membered cyclic anhydride in the C-ring. Following the B2 route, their biosynthetic precursor must be applandiketone A (**83p**), an α-dicarbonyl derivative which is also present in the mushroom [162].

As happened with other derivatives, the structure of applanhydrides A (**83**) and B (**84**) was corrected as **83c** and **84c** [136].

### 7.3. Oleananes

Compound **85** was isolated from the stems *Microtropis fokienensis* growing in Taiwan (Table 27). It was reported to be a pentacyclic triterpenoid with an oleanane skeleton, possesing in its A-ring a 7-membered cyclic anhydride formed most likely following the B2 route. Dione **85p** was also found in this species. This compound showed significant anti-inflammatory activity against superoxide anion generation and elastase release by neutrophils in response to formyl-Met-Leu-Phe/cytochalasin B, with IC_50_ values of 2.1/2.9 μM [163]. As happened previously, the structure of **85** was corrected as **85c** [136].

### 7.4. Friedelanes

In 2009, lobatanhydride (**86**) was isolated from the dried leaves of *Crossopetalum lobatum*, constituting the first example of a triterpene anhydride with an expanded A-ring [164] (Table 28). It is considered that the biosynthesis of the cyclic anhydride should follow the B1 pathway, starting from friedelin (**86p**) as a precursor.

Celastranhydride (**87**) was isolated from *Kokoona zeylanica*, *K. reflexa*, *Cassine balae* and *Reissantia indica*, and can be considered as a derivative of pristemirin, a quinone methide friedelane [165,166]. Its biosynthesis was proposed by Gamlath et al. from pristimerin (**87p**), also found in the plant. The formation of the anhydride was proposed to occur by the B1 route, starting from the overoxidation of the methyl in the A-ring of pristemerin to generate **I**, which undergoes an oxidative cleavage to produce **II**, which evolves to the final anhydride via the double decarboxylation of **III** (Figure 16) [165].

## 8. Conclusions

Terpenes containing cyclic acid anhydride in their structure have been largely disregarded, despite the biological activities they display. In this revision, more than 100 cyclic terpenic anhydrides and related compounds have been described. Different biosynthetic routes can be proposed for these natural products. These routes were supported by the occurrence in the same plant of plausible precursors. Special attention has been given to cantharidin, a norsesquiterpene produced by the blister beetle (*Coleoptera meloidae*) and false blister beetle (*Coleoptera oedemeridae*), because of the wide range of activities it presents. A plausible biosynthesis of cantharidin has been proposed.

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
