# Peer review of "Natural Occurring Terpene Cyclic Anhydrides: Biosynthetic Origin and Biological Activities"

_biomolecules, 2024, doi:10.3390/biom14080955_

Round 1

Reviewer 1 Report

Comments and Suggestions for Authors

Recommendation: Publish in The Journal of Biomolecules after minor revisions.

Comments: The authors have made a significant contribution to the field of natural products with their comprehensive review on terpenic cyclic-anhydrides. This review meticulously covers the isolation, biological activities, and biosynthetic pathways of these natural products, incorporating a thorough analysis of previous literature. The depth of information presented suggests that this work will be warmly welcomed within the natural products and organic chemistry communities.

I am pleased to recommend the acceptance of this paper for publication in the Journal of Biomolecules with minor revisions:

1.     Elaborate on the biosynthetic pathways for cantharidin, particularly focusing on the mechanism.

2.     It would be beneficial to propose a biosynthetic pathway for Chloramultiol G as it represents an intriguing synthetic target.

Author Response

Reviewer 1:

Comments: The authors have made a significant contribution to the field of natural products with their comprehensive review on terpenic cyclic-anhydrides. This review meticulously covers the isolation, biological activities, and biosynthetic pathways of these natural products, incorporating a thorough analysis of previous literature. The depth of information presented suggests that this work will be warmly welcomed within the natural products and organic chemistry communities.

I am pleased to recommend the acceptance of this paper for publication in the Journal of Biomolecules with minor revisions:

  1. Elaborate on the biosynthetic pathways for cantharidin, particularly focusing on the mechanism.
  2. It would be beneficial to propose a biosynthetic pathway for Chloramultiol G as it represents an intriguing synthetic target.

Comments 1:

According to the reviewer´s suggestion, we have added more information details to the proposed biosynthetic pathways for cantharidin. In particular, the mechanism part has been described in more detail, as suggested. These modifications have been highlighted in the text in yellow in our revised manuscript.

Comments 2:

According to the reviewer´s suggestion, we have added a proposal for a biosynthetic pathway for Chloramultiol G. This inclusion has been highlighted in the text in yellow in our revised manuscript.

Reviewer 2 Report

Comments and Suggestions for Authors

Several specific issues need addressing:

1. Manuscript Title: What is meant by 'Origen'?

2. Page 1, Line 19-20: "...such as their related imides have also been taken described due to its significant biological activity." This sentence needs careful revision.

3. Line 173: Incorrect naming of reaction: Bayer-Villiger oxidation.

4. Scheme 3: Please consider labeling the carbons mentioned from lines 180-184.

5. Font for References differ from the main text, and there is some formatting error from references numbered 100-163.

Overall, the authors should meticulously review their work for accuracy and detail, given that it is a proposed review paper. They should also ensure thorough checking of English expression, grammar, and formatting throughout the manuscript before submitting a revised version. Careful proofreading is essential, and I recommend ‘Minor Revision’.

Comments on the Quality of English Language

Overall, the authors should meticulously review their work for accuracy and detail, given that it is a proposed review paper. They should also ensure thorough checking of English expression, grammar, and formatting throughout the manuscript before submitting a revised version. Careful proofreading is essential.

Author Response

Reviewer 2:

Several specific issues need addressing:

  1. Manuscript Title: What is meant by 'Origen'?
  2. Page 1, Line 19-20: "...such as their related imides have also been taken described due to its significant biological activity." This sentence needs careful revision.
  3. Line 173: Incorrect naming of reaction: Bayer-Villiger oxidation.
  4. Scheme 3: Please consider labeling the carbons mentioned from lines 180-184.
  5. Font for References differ from the main text, and there is some formatting error from references numbeyellow 100-163.

Overall, the authors should meticulously review their work for accuracy and detail, given that it is a proposed review paper. They should also ensure thorough checking of English expression, grammar, and formatting throughout the manuscript before submitting a revised version. Careful proofreading is essential, and I recommend ‘Minor Revision’.

Response 1:

Since different biosynthetic pathways are described for the final oxidations leading to the anhydride ring, we have included in our manuscript a proposal of which of these mechanisms is involved specifically for each of the anhydrides described. Thus, by the term origin we mean the concrete pathway that is proposed to lead to each anhydride.

Response 2:

The referyellow sentence has been revised as follows: "such as their natural imide derivatives have also been described due to its significant biological activity." This modification has been highlighted in the text in yellow in our revised manuscript.

Response 3:

According to the reviewer´s suggestion, the name of reaction has been modified as: “oxidation”. This modification has been highlighted in the text in yellow in our revised manuscript.

Response 4:

As suggested by the reviewer, the carbons mentioned in Scheme 3 have been labelled. This modification has been highlighted in the text in yellow in our revised manuscript.

Response 5:

According to the reviewer´s suggestion, references and main text share the font in our revised manuscript. We have also corrected the formatting error found in the reference part, in fact, the mentioned references have been checked one by one to make sure they are correct Thes modifications have been highlighted in the text in yellow in our revised manuscript.

Comments 1:

According to the reviewer´s suggestion, we have realized a careful proofreading of our manuscript. As previously indicated, we have the mentioned references one by one to make sure they are correct. We have also revised the structures and we have corrected some errors found. Finally, the English expression and grammar has been revised by Prof M J. De la Torre from the Department of English and German Philology of the University of Granada.

Reviewer 3 Report

Comments and Suggestions for Authors

Comments of the reviewer to the masnucript:  „ Natural Occurring Terpene Cyclic-Anhydrides: Biosynthetic Or- 2 igen and Biological Activities”

The manuscipt is a huge work, introduce a comprehesive review about the natural occurring terpene cyclic-anhydrides: biosynthetic or- 2 igen and biological activities.

The work is well-organized and written. For the introduction of the topic the authors use a wide list of the references, which helped the authors to go around the topic sufficiently and to discuss it thoroughly.

Only few things that could be imporved:

I found self-citation which could be put a little more formally, it is not necessary to refer to persons in the following part of the manuscript:

(Gibelactol (65p), also isolated 765 from G. fujikuroi by one of us [141] must be an intermediate of the process, which falls 766 under the so-called route A. The plausible precursor of fujenoic acid (66p) was likewise 767 described by one of us in a mutant of G. fujikuroi [142]. Fujenal has been described to in- 768 hibit clover infection with Rhizobium trifolii [143]).

Some additional questions that could be discussed:

·         Regarding the cell proliferation activity of the discussed compunds in a whole paper only few cell lines are mentioned. What about other types of human cancer cell lines, eg. CAKI-1, CAKI-2 and A-498 human kidney cancer cell lines? If there are some information, please add.

·         Is there any in vitro experiments in the background literature that discussing the signaling pathways that could be effected by the discussed compounds?

·         The manuscript mention several time IC50, cell cytotoxic activity and concentration of the specific compounds. I would be very practical to collect cell cytotoxicity effect, IC 50 with concentration of the specific  cell lines or microorganisms in a separate table where it could be applicable.

·         Does cantharidin has any effect on any type of kidney cancer cells?

Author Response

Reviewer 3:

The manuscipt is a huge work, introduce a comprehesive review about the natural occurring terpene cyclic-anhydrides: biosynthetic or- 2 igen and biological activities.

The work is well-organized and written. For the introduction of the topic the authors use a wide list of the references, which helped the authors to go around the topic sufficiently and to discuss it thoroughly.

Only few things that could be imporved:

I found self-citation which could be put a little more formally, it is not necessary to refer to persons in the following part of the manuscript:

(Gibelactol (65p), also isolated 765 from G. fujikuroi by one of us [141] must be an intermediate of the process, which falls 766 under the so-called route A. The plausible precursor of fujenoic acid (66p) was likewise 767 described by one of us in a mutant of G. fujikuroi [142]. Fujenal has been described to in- 768 hibit clover infection with Rhizobium trifolii [143]).

Some additional questions that could be discussed:

  • Regarding the cell proliferation activity of the discussed compunds in a whole paper only few cell lines are mentioned. What about other types of human cancer cell lines, eg. CAKI-1, CAKI-2 and A-498 human kidney cancer cell lines? If there are some information, please add.
  • Is there any in vitro experiments in the background literature that discussing the signaling pathways that could be effected by the discussed compounds?
  • The manuscript mention several time IC50, cell cytotoxic activity and concentration of the specific compounds. I would be very practical to collect cell cytotoxicity effect, IC 50 with concentration of the specific  cell lines or microorganisms in a separate table where it could be applicable.
  • Does cantharidin has any effect on any type of kidney cancer cells?

Comments 1:

According to the reviewer´s suggestion, we have deleted self-citation in the paragraph indicated.

Response 1:

Following the reviewer´s indication, several cytotoxicity studies were found, but mostly limited to the action of cantharidin. Thus, a list of the cancer affected by cantharidin has bed added to the revised article, including the corresponding cell lines. Regarding the kidney cancer, it was described the cantharidin IC50 for several renal cancer lines, including TK-10, ACHN, RXF393, SN-12C, CAKI-1, 786-0, UO-31 and A498, with values ranging from 3 to 20 mM. These data have been incorporated to our revised manuscript and highlighted in the text in yellow.

Response 2:

Revision of the literature revealed than cantharidin can affect different cell signaling pathways, with MAPK, Bcl2/Bax, JNK, NF-κB, ERK, PKC, β-catenin, Wnt/β-catenin, PI3/AKT and PIk3/ATK/mTOR being recognized as its potential molecular targets. These data have been incorporated to our revised manuscript and highlighted in the text in yellow.

Response 3:

According to the reviewer´s suggestion, we started to elaborate a table including active compounds together with their cell cytotoxicity effect, IC 50 and specific cell lines. However, once cantharidin and its analogues are not included in the table for the reasons argued in Response 1, we realized that the table was limited to data from only 5 molecules, which led us to finally decide not to include the table in our revised version of our manuscript.

Response 4:

As mentioned in “Response 1”, several renal cancer lines are affected by exposure to cantharidin and norcantaridin. Specific cell lines and IC50 vales have been added to our revised manuscript.